

# A Holocene black carbon ice-core record of biomass burning in the Amazon Basin from Illimani, Bolivia

Dimitri Osmont[1,2,3], Michael Sigl[1,2], Anja Eichler[1,2], Theo M. Jenk[1,2], Margit Schwikowski[1,2,3]

[1]Laboratory of Environmental Chemistry, Paul Scherrer Institut, 5232 Villigen, Switzerland
[2]Oeschger Centre for Climate Change Research, University of Bern, 3012 Bern, Switzerland
[3]Department of Chemistry and Biochemistry, University of Bern, 3012 Bern, Switzerland

*Correspondence to*: Margit Schwikowski (margit.schwikowski@psi.ch)

**Abstract.** The Amazon Basin is one of the major contributors to global biomass burning emissions. However, regional paleofire trends remain partially unknown. Due to their proximity to the Amazon Basin, Andean ice cores are suitable to
10 reconstruct paleofire trends in South America and improve our understanding of the complex linkages between fires, climate and humans. Here we present the first refractory black carbon (rBC) ice-core record from the Andes as a proxy for biomass burning emissions in the Amazon Basin, derived from an ice core drilled at 6300 m a.s.l. from Illimani glacier in the Bolivian Andes and spanning the entire Holocene back to the last deglaciation 13000 years ago. The Illimani rBC record displays a strong seasonality with low values during the wet season and high values during the dry season due to the
15 combination of enhanced biomass burning emissions in the Amazon Basin and less precipitation at the Illimani site. Significant positive (negative) correlations were found with reanalyzed temperature (precipitation) data, respectively, for regions in Eastern Bolivia and Western Brazil characterized by a substantial fire activity. rBC long-term trends indirectly reflect regional climatic variations through changing biomass burning emissions as they show higher (lower) concentrations during warm/dry (cold/wet) periods, respectively, in line with climate variations such as the Younger Dryas, the 8.2 ka
event, the Holocene Climatic Optimum, the Medieval Warm Period or the Little Ice Age. The highest rBC concentrations of the entire record occurred during the Holocene Climatic Optimum between 7000 and 3000 BC, suggesting that this outstanding warm and dry period caused an exceptional biomass burning activity, unprecedented in the context of the past 13000 years. Recent rBC levels, rising since 1730 AD in the context of increasing temperatures and deforestation, are similar to those of the Medieval Warm Period. No decrease was observed in the 20th century, in contradiction with the global picture
("broken fire hockey stick" hypothesis).

## 1 Introduction

Fires play a major role in the global carbon cycle by emitting aerosols and greenhouse gases. Current global $CO_2$ emissions due to biomass burning represent ~50 % of those originating from fossil fuel combustion (Bowman et al., 2009). The mean annual burned area worldwide amounts to 348 Mha for the time period 1997–2011 (Giglio et al., 2013). South America is,
after Africa, the second most affected region by biomass burning, accounting for 16 to 27 % of the global annual burned area between 1997 and 2004 (Kloster et al., 2010; Schultz et al., 2008), leading to carbon emissions ranging from ~300 to 900 TgC yr$^{-1}$ (Kloster et al., 2010; Schultz et al., 2008). Biomass burning mainly occurs in Southern Hemisphere South America, representing 13.6 % of the global annual carbon emissions from biomass burning (van der Werf et al., 2010), with Brazil and Bolivia being the two countries most affected, accounting for 60 % and 10 % of active fire observations, respectively (Chen
et al., 2013). Savannas (*cerrados*) and seasonally dry tropical forests (SDTFs) located at the southern edge of the Amazon Basin are prone to extensive fires during the dry season between June and October, when many fires are ignited for land clearance purposes for agriculture and grazing (Mouillot and Field, 2005; Power et al., 2016). Savanna burned area over South America remained fairly stable over the 20th century (Mouillot and Field, 2005; Schultz et al., 2008). However, this trend masks regional discrepancies: while burning decreased along the Brazilian coast, it has strongly increased in the





western part due to deforestation (Mouillot and Field, 2005). On the contrary, tropical rain forests centered further north in the Amazon Basin rarely burn naturally due to persistent moist conditions and limited dry lightning (Bowman et al., 2011; Cochrane, 2003), but since the 1960s they have experienced intensive deforestation fires at their southern edge, known as the "arc of deforestation" (Cochrane et al., 1999). Southern Hemisphere South America accounted for 37 % of all deforestation

fires worldwide over the time period 2001–2009 (van der Werf et al., 2010). However, the contribution from deforestation fires to the total number of fires observed in South America seems to have decreased since 2005, particularly in Brazil due to stricter environmental policies, while fire activity significantly increased in Bolivia (Chen et al., 2013). A similar observation was made by van Marle et al. (2017a), who reported a strong increase in deforestation fires in the 1990s followed by a general decline since the 2000s.

For the time period before the start of satellite measurements (1980s), the lack of accurate data from the Amazon Basin hinders a detailed reconstruction of fire history. Historical reconstructions based on fire statistics, land-use practices and vegetation type and history have shown a dramatic increase of fires in the forested parts of South America over the last century (Mouillot and Field, 2005) as burning was almost absent from the Amazon Basin before the 1960s. So far, charcoal records from lake sediment cores were the only way to infer paleofire trends in this region before the 20[th] century. They

revealed that biomass burning trends in Tropical South America were less pronounced than in other regions of the Americas, underlining the absence of a clear driver for biomass burning possibly due to the large diversity of climate, vegetation and topography in this region (Power et al., 2012). Nevertheless, the last 2000 years showed an overall slight decrease in fire activity until around 1800 AD, followed by a strong increase in the 20[th] century (Power et al., 2012). This is in contradiction with the "broken fire hockey stick" hypothesis (Marlon et al., 2008) suggesting a global decoupling since 1870 AD between

the decreasing biomass burning trend and its main drivers, namely increasing temperature and population density, due to fire management and global expansion of intensive agriculture and grazing leading to landscape fragmentation. Composite charcoal records for Tropical South America display a great variability through the entire Holocene, with higher-than-present biomass burning levels in the mid-Holocene between 6500 and 4500 BP (Marlon et al., 2013). However, charcoal records only reflect local to regional conditions and therefore have to be compiled while ice cores have the potential to integrate

information over continental scales (Kehrwald et al., 2013). Ice-core records from Antarctica have also shown their ability to give an insight into past biomass burning trends in the Southern Hemisphere, revealing elevated biomass burning activity around 8000 to 6000 BP in Southern America (Arienzo et al., 2017), or confirming an overall agreement with the "broken fire hockey stick" hypothesis (Wang et al., 2010). Given the remoteness of the Antarctic continent, limitations may arise from transport patterns, thus advocating for the use of ice-core records located closer to the source regions.

As the major moisture source in the tropical Andes is the Amazon Basin and ultimately the Atlantic Ocean (Garreaud et al., 2003; Vuille et al., 2003), ice cores from tropical Andean glaciers could serve as potential archives of past biomass burning trends in the Amazon Basin and thus form the missing link between South American lake sediment charcoal records and Antarctic ice-core records, which could be helpful to better constrain fire models and historical fire databases (van Marle et al. 2017b). However, the preservation of a biomass burning signal in tropical Andean ice cores has never been extensively

investigated so far. Bonnaveira (2004) noted that an Amazonian biomass burning contribution was expressed in the concentrations of organic species (e.g. oxalate) at the end of the dry season (August–October) in aerosols collected at Plataforma Zongo, 40 km north of the Illimani site (Bolivia). The charcoal record from the Sajama ice core (Bolivia) did not suggest marked changes in biomass burning activity over the last 25000 years, except in the most recent sample due to increasing anthropogenic burning and ore-smelting (Reese et al., 2013). Nearby sedimentary charcoal records do show a

biomass burning variability in the Bolivian Amazonian lowlands through the Holocene (Brugger et al., 2016; Power et al., 2016), with enhanced burning during warmer/drier periods such as the early to mid-Holocene (approximately from 8000 to 5500 BP, Baker et al., 2001) and limited burning during colder/wetter periods such as the last deglaciation or the Little Ice Age.



Ice-core studies suggested that a variety of biomass burning proxies could be used to reconstruct paleofires. For instance, ammonium ($NH_4^+$) has been widely analyzed in polar ice cores from Greenland (Fischer et al., 2015; Legrand et al., 2016) and Antarctica (Arienzo et al., 2017). Simple organic acids (formate, oxalate) were also considered, but they can experience post-depositional effects (Legrand et al., 2016). However, the aforementioned compounds are not specific proxies as they

also reflect continuous biogenic emissions from vegetation and soils in their background variations, while only peak values can be associated with biomass burning events (Fischer et al., 2015). Black carbon (BC), produced by the incomplete combustion of biomass and fossil fuels (Bond et al., 2013), has the advantage of being a specific proxy for biomass burning in preindustrial times, when no significant anthropogenic sources existed. Aerosol source apportionment studies in Amazonia have shown that recent BC emissions in this region originate only from biomass burning (Artaxo et al., 1998).

Several ice-core studies link preindustrial BC variations with biomass burning trends (Arienzo et al., 2017; Osmont et al., 2018; Zennaro et al., 2014). However, an increasing anthropogenic BC contribution from fossil fuel combustion has been observed in ice cores from Greenland (Keegan et al., 2014; McConnell et al., 2007; Sigl et al., 2013) and the Alps (Jenk et al., 2006; Lavanchy et al., 1999; Sigl et al., in press) since the second half of the 19[th] century, and from Eastern Europe (Lim et al., 2017) and Asia (Kaspari et al., 2011; Wang et al., 2015) in the last decades.

Here, we present the first Andean BC ice-core record, derived from the analysis of the Illimani 1999 (IL-99) ice core. When referring to our measurements using the laser-induced incandescence method, the term refractory black carbon (rBC) will be employed, following the recommendations of Petzold et al. (2013). After discussing the seasonality of the rBC signal and the connections with regional climate parameters and biomass burning, we will present rBC long-term trends of the last millennium and through the Holocene, link them with climate variability and compare them to existing ice-core and lake

sediment records.

## 2 Methods

### 2.1 Ice core and site characteristics

In June 1999, two ice cores were drilled at 6300 m a.s.l. on Nevado Illimani, Bolivia, on a glacier saddle between the summits of Pico Central and Pico Sur (16°39' S, 67°47' W, Fig. 1) by a joint French-Swiss team from the Institut de

Recherche pour le Développement (IRD, France) and the Paul Scherrer Institut (PSI, Switzerland), using the Fast Electromechanical Lightweight Ice Coring System (FELICS, Ginot et al., 2002a). Bedrock was reached at 136.7 m depth (French core) and 138.7 m depth (Swiss core, this study). Low boreholes temperatures (<−7 °C) and very few ice lenses indicative of meltwater percolation ensured a good preservation of the chemical signal recorded in the ice core (Kellerhals et al., 2010a). Further details can be found in Knüsel et al. (2003) and Knüsel et al. (2005). Bonnaveira (2004) investigated

post-depositional effects such as sublimation and wind scouring and showed that their influence on the preservation of ionic species remained limited compared to actual seasonal variations in concentration.

The climate of the Bolivian Altiplano is characterized by a wet season during the Austral summer (November–March) and a dry season during the Austral winter (April–October). Moisture mainly originates from the Amazon Basin, and ultimately from the Atlantic Ocean (Garreaud et al., 2003; Vuille et al., 2003). Moreover, an interannual variability in precipitation is

35 induced by El Niño Southern Oscillation (ENSO) processes. El Niño years tend to be drier on average as they inhibit moisture influx from the East whereas La Niña years are usually wetter (Garreaud and Aceituno, 2001; Garreaud et al., 2003). The Illimani, located on the eastern margin of the Altiplano, can receive moisture influx also during the dry season, leading to a less pronounced seasonality with summer months (December–January–February,) representing only 50–60 % of the annual mean precipitation (Garreaud et al., 2003). This trend is also reflected in the Illimani ice core record, compared to

40 other Andean ice-core sites showing a more pronounced seasonality (Ginot et al., 2002b). Similarly, the precipitation

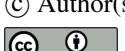



modulation by ENSO remains weaker at the Illimani site compared to the western side of the Andes (Garreaud et al., 2003; Vuille et al., 2000).

Previous studies on the IL-99 ice core include the investigation of a potential impact of ENSO on the ionic records (Knüsel et al., 2005), revealing elevated dust values during warm phases of ENSO, the use of thallium as a possible volcanic eruption

tracer (Kellerhals et al., 2010b), regional temperature reconstruction for the last 1600 years based on the $NH_4^+$ record (Kellerhals et al., 2010a), the historical reconstruction of regional silver production and recent leaded gasoline pollution derived from the lead record (Eichler et al., 2015), and that of copper metallurgy inferred from the copper record (Eichler et al., 2017), showing that earliest extensive copper metallurgy started in the Andes 2700 years ago.

### 2.2 Ice-core dating

Dating of the core was performed by using a multi-parameter approach combining annual layer counting of the electrical conductivity signal, reference horizons such as the 1963 AD nuclear fallout peak and volcanic eruptions (AD 1258, 1815, 1883, 1963, 1982, 1991), the $^{210}Pb$ decay (Knüsel et al., 2003), and $^{14}C$ dating (Kellerhals et al., 2010a). A continuous age-depth relationship was established by fitting a two-parameter glacier flow model through the reference horizons, except between the last five $^{14}C$ ages where linear interpolation was used due to the very strong layer thinning (Kellerhals et al.,

2010a), resulting in a bottom age of 13000 years BP and an overall accumulation rate of 0.58 m yr$^{-1}$ weq (water equivalent). Dating uncertainty is estimated to be ±2 years in the vicinity of volcanic horizons and ±5 years otherwise back to 1800 AD, ±20 years for the time period 1250–1800 AD and ±110 years at the youngest $^{14}C$ age (1060–1280 BP, Kellerhals et al., 2010a).

### 2.3 Sampling and rBC analysis

The IL-99 ice core was cut for rBC analysis into 1.9 x 1.9 cm sections from the inner part of the core in a −20 °C cold room at PSI following clean protocols (Eichler et al., 2000). Sampling resolution was 10 cm for the first 316 samples down to 33.15 m depth (spanning 1966–1999 AD) and 3–4 cm for the remaining 2754 samples below 33.15 m down to the bottom. A total of 3070 samples was obtained. 243 replicates from parallel ice-core sticks were cut from 12 different ice-core sections to check the reproducibility of our analyses. Furthermore, the section between 127.4 and 133 m depth (spanning roughly 0–

2000 BC) was resampled at 3–7 cm resolution (121 samples) specifically due to poor ice-core quality (chips). Samples for the rBC analyses were collected in pre-cleaned 50 mL polypropylene tubes and stored at −20 °C.

The entire IL-99 ice core was analyzed for rBC at PSI between April and June 2017, following the method described by Wendl et al. (2014). After melting the ice-core samples at room temperature and 25 min sonication in an ultrasonic bath, rBC was quantified using a Single Particle Soot Photometer (SP2, Droplet Measurement Technologies, USA, Schwarz et al.,

2006; Stephens et al., 2003) coupled to an APEX-Q jet nebulizer (Elemental Scientific Inc., USA). Further analytical details regarding calibration, reproducibility and autosampling method can be found in Osmont et al. (2018). Replicate samples for the IL-99 ice core showed good reproducibility (r = 0.65, p < 0.001, n = 243) in particular regarding rBC peak values and trends, while the resampled part in the period 0–2000 BC (121 samples) showed notable agreement with the original dataset, thus confirming the reliability of our rBC analysis.

### 3 Results and discussion

### 3.1 rBC seasonal variability in the Illimani ice core

The Illimani rBC record displays a strong seasonal variability, with high concentrations corresponding to the maximum of the dry season (June–October) and low concentrations during the wet season (November–March). In the IL-99 ice core, peak values typically range between 2 and 10 ng g$^{-1}$, with a high year-to-year variability and a maximum of 13.3 ng g$^{-1}$ in 1996,





while the wet season background remains fairly constant, below 0.5 ng g$^{-1}$ (Fig. 2a). The observed rBC seasonality is similar to previous observations made on records of trace elements (Correia et al., 2003) and major ions (Knüsel et al., 2005; see e.g. $NH_4^+$ and $Ca^{2+}$, Fig. 2b-c) reflecting the seasonality in precipitation (Fig. 2d). During the wet season, abundant precipitation occurs, which dilutes the chemical signal in the snow, whereas during the dry season, the little amount of precipitation leads

to highly concentrated wet deposition and also enables dry deposition of dust particles (Bonnaveira, 2004; Correia et al., 2003; De Angelis et al., 2003). The seasonal signal in the Illimani ice core is therefore mainly the result of transport to and deposition at the Illimani site combined with the fact that dust mobilization from the Altiplano (Kellerhals et al., 2010a; Knüsel et al., 2005) and biomass burning emissions in the Amazon Basin also peak during the dry season (Mouillot and Field, 2005; Power et al., 2016).

**3.2 Connection with climate parameters in South America during the 20$^{th}$ century**

During the last century, rBC concentrations did not show an evident long-term trend, but decadal changes peaking in the 1900s, 1940s and 1960s (Fig. 3a). These maxima are not in agreement with model-based BC emissions (Fig. 3c). We extracted the time series of BC emissions from biomass burning for the 5x5° grid cell containing the Illimani site used in the Coupled Model Intercomparison Project Phase 6 (CMIP6) simulations (van Marle et al., 2017b) and compared it to the IL-99

rBC record for the time period 1900–2000 AD. Even if all the BC emissions recorded at Illimani are not expected to come solely from this grid cell, it is striking to note that estimated BC emissions remained perfectly constant until the start of satellite measurements in the 1980s, when the data coverage greatly improved. Estimated BC emissions subsequently exhibited much more variability and increased by more than one order of magnitude until the late 1990s.

A direct relationship between the rBC record and biomass burning trends in the Amazon Basin before 1999 cannot be

assessed due to the lack of accurate biomass burning statistics from Bolivia and Brazil, where the number of active fires is retrieved from satellite data starting only in 1998. In addition, data about the burned area remain scarce and are associated with larger uncertainties, despite their greater significance in terms of environmental impacts and aerosol emissions (Montellano, 2012).

For investigating major causes for rBC changes during the 20$^{th}$ century, we studied spatial and temporal correlations between

the IL-99 rBC record and two important drivers of biomass burning activity, namely temperature and precipitation. Significant correlations ($p < 0.05$) between the IL-99 rBC record and re-analyzed temperature and precipitation from the NCEP/NCAR R1 dataset were found for areas in the Amazon Basin located East of the Illimani site, which are assumed to be the main source regions of the rBC deposited at Illimani (Fig. 4). 5-year moving averages were used owing to the ice-core dating uncertainty which prevents detection of an annual connection with temperature and precipitation data. Positive

correlations with temperature are highest along the arc of deforestation in Brazil and in regions of Eastern Bolivia (states of El Beni and Santa Cruz) and Western Brazil (state of Rondônia and Mato Grosso) where extensive fires occur during the dry season. Similarly, negative correlations with precipitation are highest along the Bolivian-Brazilian border, for the states of Santa Cruz and Mato Grosso. Comparisons between temperature/precipitation time series for the Amazon Basin (defined here as the region between 4 °N–16 °S and 76 °W–51 °W) and the IL-99 rBC record confirm that higher rBC concentrations

are observed during warmer and drier periods, such as the 1900s, the 1940s and the 1960s (Fig. 3d and 3e). Different temperature/precipitation datasets were used to highlight their strong variability, but the main conclusion remains unchanged. Depending on the used dataset, variations in temperature and precipitation account for 18–64 % and 1–18 % of the rBC variance, respectively. However, the correlation between the IL-99 rBC record and precipitation datasets is never significant at the 0.05 level, suggesting a predominant influence of temperature on regional fire activity.

Potential connections between the IL-99 rBC record and the ENSO phenomenon were also investigated. In general, and similarly to the Altiplano, El Niño phases of ENSO induce drier and warmer conditions over the Amazon Basin, while La Niña phases are wetter and cooler (Aceituno, 1988; Foley et al., 2002; Garreaud et al., 2009). The trend is more pronounced



during the wet season (Garreaud et al., 2009). However, this relationship weakens towards the western part of the Amazon Basin (Garreaud et al., 2009; Ronchail et al., 2002) and becomes more complex on the Bolivian slopes between the Amazon Basin and the Altiplano as opposite effects can be observed depending on the altitude (Ronchail and Gallaire, 2006). To determine whether ENSO can modulate rBC concentrations in the Illimani ice core, we compared the 20th-century IL-99 rBC

record to the Multivariate ENSO Index (MEI, Fig. 3b) spanning 1950–2018 (Wolter and Timlin, 1993, 1998) and the Extended MEI reaching back to 1871 (Wolter and Timlin, 2011). The low correlation coefficient between the rBC record and the MEI indicates no evident impact of ENSO. Interestingly, the highest two rBC annual values occurred during some of the most outstanding El Niño events (1905–1906 and 1941), but rBC values can also remain low during strong El Niño phases, for instance in 1929–1930. Conversely, rBC annual values are not necessarily low during intense La Niña phases, as

seen in 1910, 1917 or 1954–1956. To comprehensively assess this relationship, we calculated the average rBC concentration for all the El Niño and La Niña years for the time period 1900–1998. Average rBC concentrations of $0.85 \pm 0.44$ ng g$^{-1}$ for El Niño years (50 years in total) and $0.93 \pm 0.42$ ng g$^{-1}$ for La Niña years (49 years in total) show that no significant difference is visible between the warm and cold phases of ENSO in the rBC record. Several hypotheses contribute to explain this lack of relationship despite drier conditions during El Niño years. First, it is well-known that the eastern side of the Andes is less

influenced by ENSO modulation as the major moisture source is the Amazon Basin and not the Pacific Ocean, contrary to the western part of the Andes (Garreaud et al., 2003; Vuille et al., 2000). Second, there is a difference in timing as the precipitation suppression induced by ENSO is more important during the wet season, whereas rBC emissions peak during the dry season due to biomass burning and limited but highly concentrated precipitation. Furthermore, if no precipitation occurs during the dry season owing to the El Niño phase of ENSO, (almost) no rBC will be deposited on the snow surface at

the Illimani site as BC is preferentially removed from the atmosphere by wet deposition (Cape et al., 2012; Ruppel et al., 2017). Lastly, as the moisture influx from the east tends to be reduced during El Niño years, the contribution from eastern-origin rBC-enriched precipitation due to biomass burning in the Amazon Basin to the total amount of precipitation at Illimani becomes weaker.

### 3.3 rBC variability over the last 1000 years

In Fig. 5a, we present the IL-99 rBC long-term record for the last 1000 years. Higher rBC concentrations were observed between 1000 and 1300 AD (average $\pm 1\sigma$ unless otherwise stated: $0.94 \pm 0.56$ ng g$^{-1}$), in agreement with the temperature maximum corresponding to the Medieval Warm Period (MWP) in the Northern Hemisphere (NH) and also previously described in the IL-99 ammonium record by Kellerhals et al. (2010a). Following the MWP, rBC concentrations slowly declined until they reached a minimum in the 18th century (average: $0.37 \pm 0.34$ ng g$^{-1}$) reflecting the Little Ice Age (LIA).

The lowest rBC concentrations were recorded around 1730 AD, following the Maunder solar minimum. The LIA decline was interrupted by higher rBC concentrations during the time periods 1460–1550 AD and 1630–1670 AD, potentially related to the apogee of the Inca Empire at the end of the 15th century and the Spanish colonization of Bolivia that started around 1535. Several cities were created on the Altiplano at that time (Sucre in 1538, Potosí in 1546 and La Paz in 1548). Mining activities rapidly took off and left an imprint on the IL-99 lead and copper records (Eichler et al., 2015, 2017, respectively).

These time periods are also corroborated by the composite charcoal record for Tropical South America showing local maxima of fire activity (Fig. 5e; Power et al., 2012). Following the 1730 AD minimum, rBC concentrations started to rise until present time (average 1900–1999: $0.91 \pm 1.23$ ng g$^{-1}$). The IL-99 cerium record (Fig. 5b; Eichler et al., 2015), used as a dust deposition tracer, shows that the MWP was characterized by dustier and drier conditions which can indirectly explain the corresponding rBC peak as dry conditions favor biomass burning and lead to reduced but more concentrated wet

deposition. Contrariwise, the LIA is generally marked by less dry conditions, except the dusty period ~1600–1650 AD. The similar long-term variability of rBC and temperature (Fig. 5a) provides evidence that temperature is indeed a major driving force for changing biomass burning activity, in agreement with the results for the 20th century (see section 3.2).





Discrepancies between the two records, particularly between 1400 and 1700 AD when temperature anomalies were constantly negative while rBC concentrations displayed higher values during 1460–1550 AD and 1630–1670 AD, can most probably be related to an additional anthropogenic impact as discussed above. To summarize, rBC concentrations in the Illimani record tend to be lower during periods of colder/wetter climate and higher during periods of warmer/drier climate,

suggesting that rBC could be used as an indirect temperature/moisture proxy through biomass burning variations.

Comparable long-term trends were found in the B40 rBC and $NH_4^+$ ice-core records from Antarctica (Fig. 5d) with lower (higher) rBC and $NH_4^+$ concentrations during the LIA (MWP), respectively (Arienzo et al., 2017). The authors of this study suggested South American biomass burning as the main source of Antarctic rBC and $NH_4^+$ throughout the Holocene, with little modification induced by long-range transport. However, some interesting differences can be noted. First, absolute rBC

concentrations are lower in the Antarctic ice cores due to the remoteness from the main source regions. Second, while the timing of the MWP matches well between the two records, the transition towards the LIA was more abrupt in the B40 record, and the LIA signal displayed less variability in the Antarctic records compared to Illimani one. Third, while the B40 $NH_4^+$ record did show an increase since 1900 AD, no clear increasing trend was visible in the last 250 years in the Antarctic rBC record, contrary to Illimani, thus highlighting the importance of considering transport processes when discussing rBC

records from the remote Antarctic regions. On the contrary, carbon monoxide (CO) ice-core records from Antarctica representative of Southern Hemisphere biomass burning follow a trend similar to the IL-99 rBC record, with a decreasing trend between 1300 and 1600 AD, a minimum in the 17[th] century and an increase for the time period 1700–1900 AD (Wang et al., 2010). Divergences between rBC and CO Antarctic records might result from their different atmospheric lifetimes implying a different spatial representativeness. rBC has a relatively short atmospheric lifetime, from 3 to 10 days (Bond et

al., 2013) while CO can remain in the atmosphere for weeks to months or even more than a year at the winter poles (Holloway et al., 2000).

In rBC ice-core records from the Arctic and Europe, a predominant anthropogenic contribution starting in the second half of the 19[th] century due to rising fossil fuel emissions was observed (Keegan et al., 2014; McConnell et al., 2007; Osmont et al., 2018; Sigl et al., 2013, in press). At Illimani, rBC concentrations follow a long-term increasing trend since the 1730 AD

minimum. It is therefore important to investigate to which extent climate variations and human activities influenced this increase. The IL-99 dust record (Fig. 5b) reveals that this increase was not driven by dustier and drier conditions leading to enhanced deposition as cerium concentrations remain low. On the contrary, temperatures likewise increase since 1720 AD, suggesting that the rBC increase is primarily driven by increasing temperatures responsible of enhanced biomass burning. The anthropogenic pollution pattern recorded in the IL-99 ice core by Pb and $NO_3^-$ (Eichler et al., 2015) shows a dramatic

increase only in second half of the 20[th] century, mainly due to emissions from traffic (Fig. 5c). Since this strong rise is not visible in the rBC record of the past 50 years, we assume that anthropogenic BC emissions from fossil fuel combustion remain minor compared to biomass-burning-related BC emissions. Charcoal composite records for Tropical South America (Fig. 5e) also show a strong increase in fire activity during the 20[th] century, explained by enhanced deforestation (Power et al., 2012). Thus, we cannot exclude that, in the 20[th] century, a certain fraction of the biomass burning increase reflected by

the rBC record does not only originate from rising temperatures but could be the result of the expansion of deforestation. However, a detailed assessment of the relative impact of those two factors cannot be obtained given the lack of accurate statistics before the satellite measurement era. The Food and Agriculture Organization (FAO) of the United Nations is monitoring forested areas since 1947 but many inconsistencies occurred in the first reports due to high uncertainties in estimating forested areas in remote regions and changing definitions and methodologies between the subsequent reports, thus

making extensive comparison impossible (Steininger et al., 2001).



### 3.4 Evidence of a Holocene Climatic Optimum dry period

The relationship between rBC concentrations and regional temperature/moisture variations extends further back in time through the entire Holocene. The bottommost part of the IL-99 ice core, between 11000 and 10000 BC, shows low rBC concentrations (Fig. 6a) as well as low $\delta^{18}$O values (Fig. 6b), indicative of a cold and wet climate corresponding to the

Younger Dryas (YD) cold period in the NH. Wet and cold conditions over the Bolivian Altiplano were suggested by an overflowing Lake Titicaca between 11000 and 9500 BC (Baker et al., 2001) and Late Glacial glacier advances in the Cordillera Real between 11000 and 9000 BC during the "Coipasa" humid phase (Zech et al., 2007), showing that the Younger Dryas was not dry on a global scale despite increasing dustiness in Greenland ice-core records (Mayewski et al., 1993). The establishment of warmer and drier conditions corresponding to the onset of the Holocene then occurred between

10000 and 9000 BC as evidenced by a pronounced increase (+5.5 ‰) in the $\delta^{18}$O record (Sigl et al., 2009), comparable to the +5.4 ‰ increase in the nearby Sajama ice core (Thompson et al., 1998), followed by a stabilization around −16 ‰ between 9000 and 7000 BC.

Around 7000 BC, warm and dry conditions abruptly prevailed as indicated by a further increase in $\delta^{18}$O (+2 ‰). These conditions that lasted until 3500 BC correspond to the Holocene Climatic Optimum (HCO). The HCO period is marked by

the highest rBC concentrations of the whole 13000-year record (rBC average 7000–3500 BC: 2.97 ± 1.77 ng g$^{-1}$). A lower accumulation rate, induced by drier conditions, might partially explain higher rBC concentrations. However it cannot be the only driving force as the highest rBC concentrations were recorded between 7000 and 6000 BC and not between 6000 and 3000 BC, despite a three-time lower accumulation rate in the second time period. Several other studies have already shown evidence of a dry HCO in Bolivia although timings might slightly differ between regions. The lowest Lake Titicaca level

(Fig. 6c) occurred between 6000 and 3500 BC in a context of maximal aridity over the Bolivian Altiplano (Baker et al., 2001). The charcoal record from Lake Titicaca (Fig. 6d) shows a broader maximum from 10000 to 1000 BC (Paduano et al., 2003). Pollen data suggests a dry period lasting from 7000 to 1100 BC and peaking between 4000 and 2000 BC. Fire activity in the Bolivian lowlands, inferred from lake-sediment charcoal records, was high between 6100 and 3800 BC in the Llanos de Moxos (Fig. 6d, Lake Rogaguado, Brugger et al., 2016), between 6000 and 5000 BC in the Chiquitano SDTF (Fig. 6e,

Laguna La Gaiba, Power et al., 2016) and between 8000 and 4000 BC around Lake Santa Rosa (Fig. 6e, Urrego, 2006). Reduced pollen concentrations in the Sajama ice core between 6000 and 3000 BC are also indicative of a drier climate (Reese et al., 2013). Composite charcoal records for tropical South America (Fig. 6b) show elevated biomass burning levels between 6000 and 2500 BC (Marlon et al., 2013), but cover a much larger area which is not only representative of the Illimani source region. In the West Antarctic Ice Sheet Divide (WD) ice core, the highest rBC deposition occurred during the

mid-Holocene from 6000 to 4000 BC (Fig. 6c, Arienzo et al., 2017). The HCO maximum appears broader than in the IL-99 ice core, probably due to a larger catchment area and the influence of long-range transport processes.

Opposite hydroclimate variations have been detected in the northern South American tropics, as shown by the titanium and iron records from the Cariaco Basin, Venezuela (Haug et al., 2001). Dry conditions prevailed during the YD while the HCO, dated between 8500 and 3400 BC, experienced the wettest conditions of the last 14000 years in this region. The Late

Holocene was then characterized by a return to a drier climate. Similarly, wetter (drier) conditions were observed during the MWP (LIA), respectively. This anti-phasing between the northern and southern South American tropics has been best explained by latitudinal variations of the Intertropical Convergence Zone (ITCZ) (Haug et al., 2001; Arienzo et al., 2017). During the HCO, in a context of a low Austral summer insolation due to orbital forcing, the ICTZ was shifted north, leading to more precipitation in the Cariaco region and to a weaker South American Summer Monsoon (SASM) responsible of drier

conditions in the southern South American tropics, as evidenced in the Illimani record. Towards the Late Holocene, increasing (decreasing) insolation seasonality in the Southern (Northern) Hemisphere, respectively, may have resulted in a progressive southward shift of the ITCZ and a strengthening of the SASM, inducing wetter conditions over southern South



American tropics but drier conditions in the northern South American tropics. A similar conclusion can be drawn for the LIA (Arienzo et al., 2017).

Throughout the last 4000 years, rBC concentrations in the IL-99 ice core showed a much lower level (Fig. 6a). Between 2250 BC and 100 AD, rBC concentrations remained low (average: $0.60 \pm 0.37$ ng g$^{-1}$) except a peak value of 1.55 ng g$^{-1}$ around 300 BC. Brugger et al. (2016) also observed a minimum of burning in a lake sediment charcoal record from the Bolivian Amazonian lowlands around 2000 BP (Fig. 6d) in response to forest extension due to increased moisture availability. After 100 AD, rBC concentrations started to rise and remained higher for the time period 150–650 AD (average: $0.86 \pm 0.32$ ng g$^{-1}$) but declined again and stayed low between 750 and 1000 AD (average: $0.65 \pm 0.41$ ng g$^{-1}$).

In addition to the long-term trends described above, a particular event in the rBC IL-99 record is of special interest. Around 6000 BC, a dip in the $\delta^{18}$O record (Fig. 6b) suggests an abrupt centennial-scale return to cooler and wetter conditions, potentially related to the NH 8.2 ka cold event detected in Greenland ice cores (Alley et al., 1997; Thomas et al., 2007), and revealing that its impacts were also apparent in southern South American tropics and not only in the North Atlantic region. According to the current consensus, this cooling was caused by the Laurentide ice cap collapse that generated enormous freshwater fluxes into the North Atlantic Ocean (Matero et al., 2017). A concurrent drop in the rBC concentration ~6000 BC (Fig. 6a) suggests that this climate anomaly also led to reduced levels of biomass burning.

## 4 Conclusions

Refractory black carbon (rBC) was analyzed in an ice core from Illimani (Bolivian Andes) spanning the entire Holocene back to the last deglaciation 13000 years ago. The high-resolution signal in the upper part of the ice cores revealed a strong seasonal pattern for rBC, with peak values during the dry season and low concentrations during the wet season as a result of the seasonality of emission sources and precipitation. Significant correlations were found between the 20$^{th}$ century rBC record and reanalyzed temperature/precipitation datasets from the Amazon Basin, particularly with regions located in Eastern Bolivia and Western Brazil experiencing high levels of biomass burning. The modulation of the seasonality by ENSO processes was shown to be weak due to the site location in the Eastern part of the Andes. The long-term rBC record was shown to behave like an indirect regional temperature/moisture proxy through biomass burning variations, with low values during cold/wet periods such as the Younger Dryas and the Little Ice Age and higher concentrations during warm/dry periods such as the Holocene Climatic Optimum and the Medieval Warm Period. These findings are supported by an array of regional paleoclimate reconstructions and by Antarctic rBC ice-core records thought to represent South American biomass burning emissions, and are primarily controlled by insolation-driven latitudinal changes of the Intertropical Convergence Zone. Evidence of a cold/wet reversal induced by the 8.2 ka event was detected in the Illimani ice core. Our work confirms that most of the Northern Hemisphere climate variations throughout the Holocene also left an imprint in the tropical Andes and that opposite hydroclimate variations were observed between northern and southern South American tropics. Lastly, the rise in rBC concentrations since 1730 AD seems only driven by increased biomass burning levels due to higher temperatures and more intensive deforestation in the last decades but does not relate to fossil fuel rBC emissions. Therefore, the "broken fire hockey stick" trend was not observed in the Illimani record. Such ice-core records are of prime importance as the current global warming endangers the preservation of these glacial archives. Glaciers in the tropical Andes have retreated at a critical rate in the last 50 years (Rabatel et al., 2013; Vuille et al., 2008), affecting drinking water supply for millions of people (Bradley et al., 2006; Kaser et al., 2010; Vergara et al., 2007) and creating new glacial lakes threatening local populations (Carey, 2005, 2012).





## Author contribution

D.O. sampled the Illimani ice cores, carried out SP2 measurements, analyzed the data and wrote the manuscript. Mi.S helped with ice cutting, rBC analyses and data interpretation. A.E. and T.M.J assisted with the data interpretation. Ma.S. designed and led the project and supervised the manuscript writing.

## Competing interests

The authors declare that they have no conflict of interest.

## Acknowledgements

We acknowledge funding from the Swiss National Science Foundation through the Sinergia project "Paleo fires from high-alpine ice cores (CRSII2_154450/1). The authors would like to thank Susanne Haselbeck for helping with the SP2

measurements; Sabina Brütsch for ion chromatography and water stable isotope analyses; Thomas Kellerhals for his previous work on the Illimani ice core; Robin Modini and Jinfeng Yuan for their technical assistance in calibrating the SP2; Sandra Brugger, Jennifer Marlon and Mitchell Power for the charcoal data; the team members of the 1999 and 2015 drilling campaigns at Illimani.

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



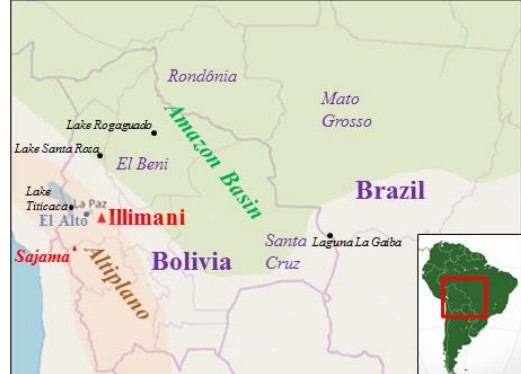

**Figure 1: Map showing the location of Nevado Illimani in Bolivia and the other sites and regions of interest in Bolivia and Brazil mentioned in the study (adapted from openstreetmap.org).**





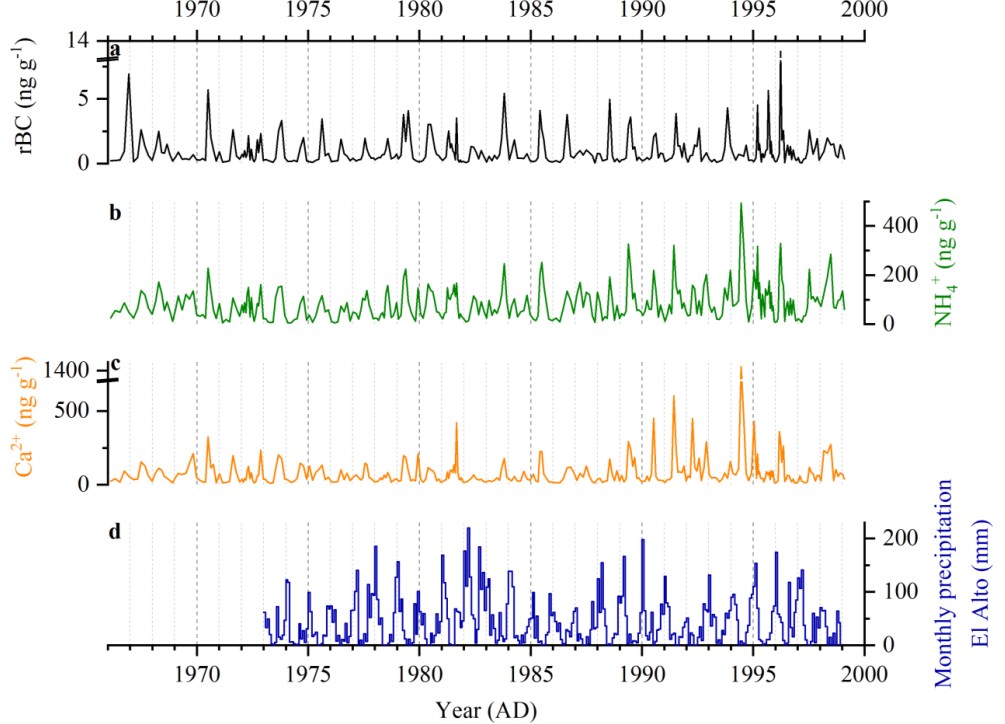

**Figure 2: Concentrations of a) rBC, b) ammonium and c) calcium in the upper 33.15 m of the IL-99 ice core (raw data). d) Monthly precipitation data from El Alto weather station, located 40 km west of Illimani, near the city of La Paz. Data is available on the website of the US National Climatic Data Center (NCDC) at the following address:**
5 **https://www7.ncdc.noaa.gov/CDO/cdoselect.cmd?datasetabbv=GSOD.**







**Figure 3: Comparison of the IL-99 rBC record with South American climate parameters for the time period 1900–2000 AD. a) rBC record from the IL-99 ice core (thin lines: annual averages, thick lines: 5-year moving averages). b) Multivariate ENSO Index (MEI, thin lines: annual averages, thick lines: 5-year moving averages, Wolter and Timlin, 1993, 1998, 2011), a higher (lower) value standing for a stronger El Niño (La Niña) event, respectively. The Extended MEI is used before 1950. c) Annual BC emissions derived from the CMIP6 simulations for the 5x5° grid cell containing the Illimani site (van Marle et al., 2017b). Comparison between the IL-99 rBC record and four d) temperature and e) precipitation datasets for the Amazon Basin (4 °N–16 °S and 51–76 °W). Data are 5-year moving averages and were extracted from the KNMI Climate Explorer. Anomalies are relative to the years 1971–2010. Pearson correlation coefficients between the IL-99 rBC record and the associated climate datasets were calculated based on 5-year moving averages and coefficients in bold are statistically significant at the 0.05 level.**



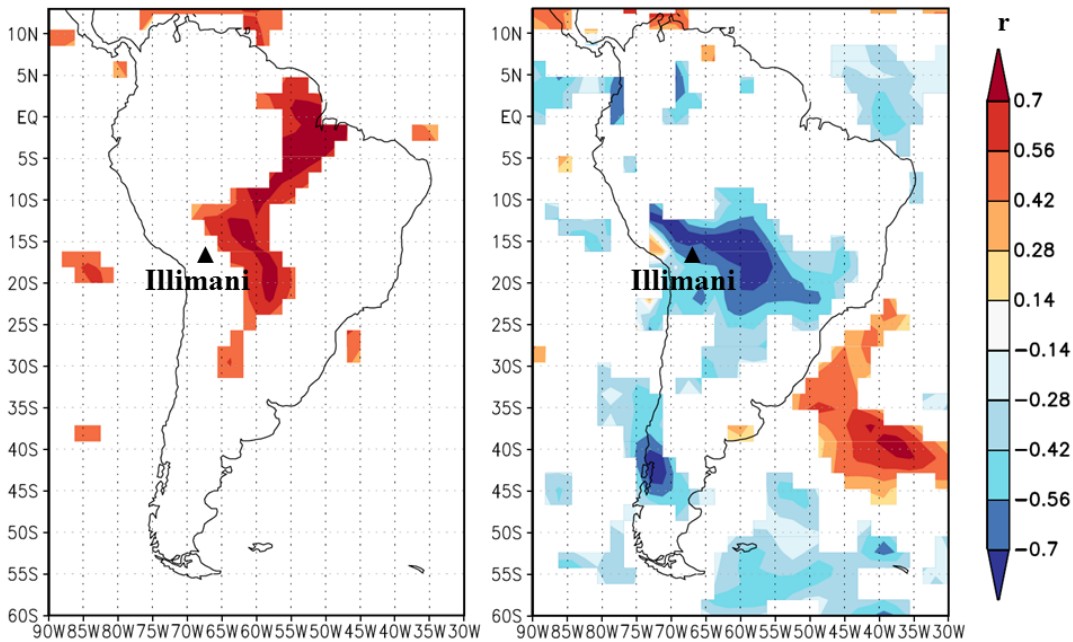

**Figure 4: Spatial correlation over South America between the IL-99 rBC record and re-analyzed temperature (left panel) and precipitation (right panel) data from the NCEP/NCAR R1 dataset for the time period 1948–1998, available on the KNMI Climate Explorer (https://climexp.knmi.nl/start.cgi). Data are annual averages smoothed with a 5-year running mean.**





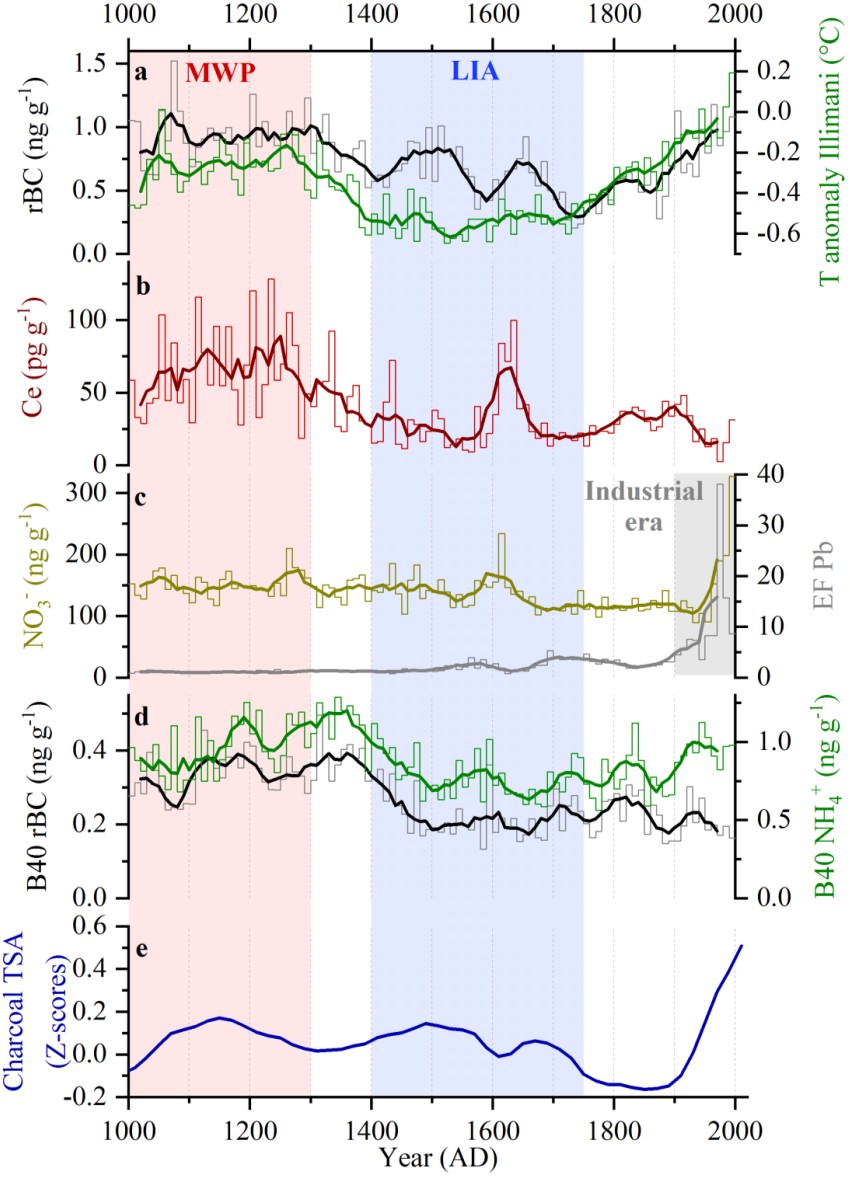

**Figure 5: Comparison for the time period 1000–2000 AD between a) IL-99 rBC record (left scale) and temperature anomalies (right scale) inferred from the ammonium IL-99 record (Kellerhals et al., 2010a), b) IL-99 cerium record as a dust proxy (Eichler et al., 2015), c) IL-99 nitrate (left scale) and lead enrichment factors (right scale) to illustrate 20th century anthropogenic impact (Eichler et al., 2015), d) Antarctic B40 rBC (left scale) and ammonium (right scale) records (Arienzo et al., 2017) and e) composite charcoal record (Z-scores of transformed charcoal influx) for Tropical South America (Power et al., 2012). Thin lines are 10-year averages and thick lines are 50-year moving averages, except for panel e where they represent 20-year averages. Timings of MWP (1000–1300 AD) and LIA (1400–1750 AD) are defined based on the IL-99 rBC and temperature records.**





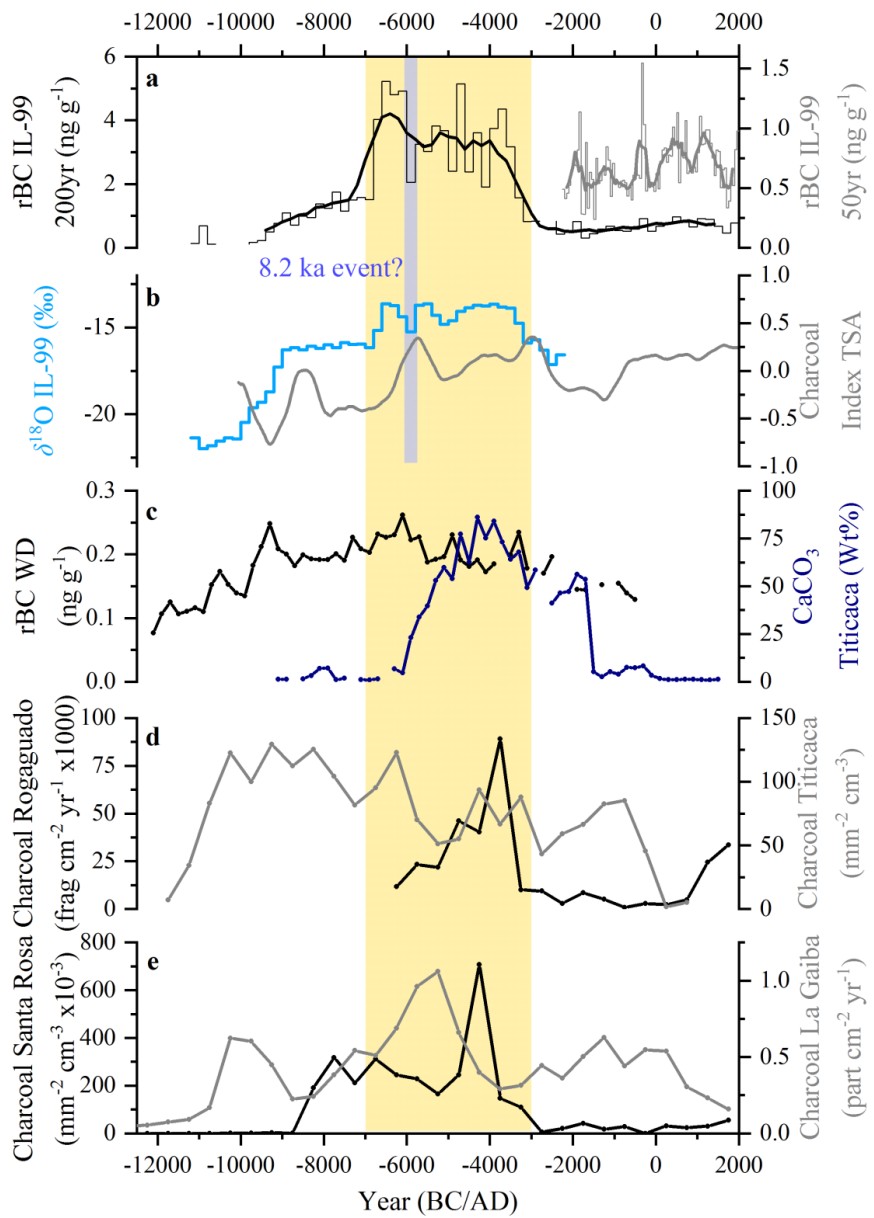

**Figure 6: Evidence of a dry period corresponding to the Holocene Climatic Optimum (HCO, yellow bar) in the IL-99 ice core and in other paleoclimatic reconstructions. a) IL-99 rBC record, 200-year averages (left scale) and 50-year averages (right scale). Thick lines are 5-point moving averages. The blue bar represents the cold/wet reversal potentially related to the 8.2 ka event. Gaps in the IL-99 rBC record are due to lack of available ice-core material due to previous samplings. b) IL-99 $\delta^{18}O$ record (Sigl et al., 2009; left scale) and composite charcoal record for Tropical South America (Marlon et al., 2013; right scale, 20-year averages with a 500-year smoothing). c) rBC record from the West Antarctic Ice Sheet Divide (WD) ice core, Antarctica (Arienzo et al., 2017; left scale) and calcium carbonate weight percent ($CaCO_3$ Wt%, right scale) recorded in a sediment core from Lake Titicaca as a proxy for lake-level variations, a higher percentage standing for a lower lake level induced by drier conditions (Baker et al., 2001). Data are 200-year averages. d) Charcoal records from Lake Rogaguado (Brugger et al. 2016; left scale) and Lake Titicaca (Paduano et al., 2003; right scale). e) Charcoal records from Lake Santa Rosa (Urrego, 2006; left scale) and Laguna La Gaiba (Power et al., 2016; right scale). Data in panels d-e are 500-year averages.**