# Peer review of "A Holocene black carbon ice-core record of biomass burning in the Amazon Basin from Illimani, Bolivia"

_Climate of the Past, 2018_

## Referee Comment (RC1) · Anonymous Referee #1 · 7 Dec 2018

"A Holocene black carbon ice-core record of biomass burning in the Amazon Basin from Illimani, Bolivia" presents a fire record from location and across a time frame where this information is sorely needed. This biomass burning record is an important contribution to fire science and to paleoclimatology. The well-written paper is within the scope of Climate of the Past, clearly presents the methodology and assumptions, and contains high-quality figures. It is evident that the authors carefully examined all aspects of the paper before submission, rather than trying to rush a draft through to submission, where the current presentation is well structured and clear. However, the authors should address the following concerns before publication:

[Figure]

The authors assume (Page 3, Lines 8-9) that "Aerosol source apportionment studies in Amazonia have shown that recent BC emissions in this region originate only from biomass burning (Artaxo et al., 1998)". This previous statement pertains to Amazonia, and does not include the major urban areas of La Paz and El Alto which are located only 10s of kilometers from Illimani. The recent article "Black carbon emission and transport mechanisms to the free troposphere at the La Paz/El Alto (Bolivia) metropolitan area based on the Day of Census (2012)" clearly demonstrates the production of traffic-related BC and deposition on regional mountains (Wiedensohler et al., 2018). I realize that this article was published recently, and that the authors may not have known of its existence when they were writing their manuscript. However, as some of the authors of the Wiedensohler et al., 2018 paper collaborate with the authors of this Climate of the Past submission, it seems likely that the authors of the submitted paper may have known about this major source of BC. Previously-published literature also clearly demonstrates the effects of BC from urban centers on Andean glaciers (Schmitt et al., 2015). Although Schmitt et al. (2015) investigate glaciers in the Cordillera Blanca, these Peruvian glaciers still have the same main moisture source of the Amazon (and by extension, of the Atlantic) as Illimani. Molina et al. (2015 and references therein) mention that 50% of the BC in the Andean region is from biomass burning, which leaves the other 50% to be produced by other sources, of which fossil fuel burning (e.g. diesel-powered vehicles) is a major component. Therefore, the assumption that BC in Illimani after industrialization is does not accurately portray regional emissions.

The authors carry the above assumption through all of their work in section 3.2, and in section 3.3, page 7, lines 22-23. The authors do account for the possibility of a fossil fuel source for the BC from ~1730 AD onwards (Page 7, Lines 24-37) including comparing their results to lead and nitrate concentrations in the IL-99 core from Eichler et al., 2015 where these concentrations reflect motor vehicle emissions. Comparing the rBC concentrations with the lead and nitrate records is a good idea, however, such a direct comparison may not be applicable. The authors only visually compare these

records (Figure 5; second paragraph on page 7) and do not numerically investigate any correlations. The 10-year averages of lead drop after the switch to unleaded gasoline, while the 10-year nitrate averages continue to climb, which is similar to the increase in rBC (Figure 5). The sampling resolution is 10-cm samples, so in this uppermost section of the core, it should be possible to examine the data at a higher resolution than 10-year averages (as nicely demonstrated in Figure 2). What is the relationship between lead, nitrate and rBC over the past century when examined at a higher resolution? (As 10-year fixed averages are a rather arbitrary number, the decadal cut-off points may introduce errors when comparing the three records. E.g. one high value can inordinately influence an entire decadal mean). What happens if you compare the records with higher resolution moving averages from 1750 AD onward? If 50% of the BC is from biomass burning, then this source would moderate the rBC concentrations where you would not expect to see an equal rise in rBC as in lead and nitrate. Therefore, the conclusion (Page 9, lines 34 and 35) "Lastly, the rise in rBC concentrations since 1730 AM seems only driven by increased biomass burning levels due to higher temperatures and more intensive deforestation in the last decades but does not relate to fossil fuel rBC emissions" is misleading.

Molina, L.T., Gallardo, L., Andrade, M., Baumgardner, D., Borbor-Córdova, M., Bórquez, R., Casassa, G., Cereceda-Balic, F., Dawidowski, L., Garreaud, R., Huneeus, N., Lambert, F., McCarty, J.L., Mc Phee, J., Mena-Carrasco, M., Raga, G.B., Schmitt, C., Schwarz, J.P., 2015. Pollution and its impacts on the South American cryosphere. Earth's Future, 3, 345-369, http://dx.doi.org/10.1002/2015EF000311.

Schmitt, C.G., All, J.D., Schwarz, J.P., Arnott, W.P., Cole, R.J., Lapham, E., Celestian, A., 2015. Measurements of light-absorbing particles on the glaciers in the Cordillera Blanca, Peru. Cryosphere 9, 331–340

Wiedensholer, A., Andrade, M., Weinhold, K., Mueller, T., Birmili, W., Velarde, F., Moreno, I., Forno, R., Sanchez, MF., Laj, P., Whiteman, D.N., Krejci, R., Sellegri, K., Reichler, T. (2018) Black carbon emission and transport mechanisms

to the free troposphere at the La Paz/El Alto (Bolivia) metropolitan area based on the Day of Census (2012). Atmospheric Environment, 194, 158-169, DOI: https://doi.org/10.1016/j.atmosenv.2018.09.032

Page 8, Lines 5 and 6: Higher lake levels in Lake Titicaca would certainly indicate a wetter climate, but why do you infer that these higher lake levels also depict a cooler climate? Due to less evaporation? The lake is also not "overflowing" when it has higher lake levels than present. Yes, the shorelines are higher than present, but the lake did not create continuous catastrophic floods.

Figures 1 and 6: I realize that there are few paleofire records for this region and so it is difficult to find records with which to compare your rBC results. (This lack of records does nicely increase the value of your work). However, as charcoal only records local to semi-regional fires, Laguna La Gaiba is too far away to provide a useful comparison with the Illimani ice core record. Illimani is more of a regional record than any lake, but comparisons between individual records are limited by the spatial record of any individual record. The four lakes contain three different ecosystems (llanos, Amazon rain forest, and seasonally dry tropical forest) and so your decision to keep these records separate rather than making a composite record makes sense. However, the current comparison with Laguna La Gaiba is beyond the geographic limits of charcoal.

Miscellaneous:

Please use a comma in numbers with five places or more.

Line 1, Page 12: Place "the" before "Illimani".

Page 1, Line 22: Omit "an" before "exceptional biomass burning activity".

Page 1, Line 24: Place "in fire activity" or "in biomass burning" after "decrease". In the previous sentence you mention both increasing temperatures and deforestation. Therefore, it is necessary to re-define what is decreasing in the following sentence.

Page 3, Line 27: Change "boreholes" to "borehole". Even if you have multiple boreholes, the plural is already included in the word "temperatures".

Page 3, Line 27: Remove "the" before "Illimani".

Page 3, Line 27: Change "can receive moisture influx also during the dry season" to "can also receive moisture influx during the dry season".

Page 5, Line 11: Change "peaking" to "peak"

Page 5, Line 12: Change "are not in agreement with" to "do not agree with"

Page 5, Line 21: Change "starting only" to "only starting"

Page 5, Line 27: Change "East" to "east"

Page 6, Line 13: Omit the word "explain" as the explanation is implicit in the word "contribute"

Page 6, line 16: Change "western part of the Andes" to "western Andes"

Page 6, Line 26: Replace "average" with "mean"

Page 7, Line 28: Replace "of" with "for"

Page 7, Line 30: Place "the" before "second half"

Page 7, Line 32: Change "charcoal composite records" to "composite charcoal records"

Page 7, Line 35: Change "could be" to "could also be"

Page 8, Line 39: Change "responsible of drier conditions" to "responsible for drier conditions"

Page 9, Line 23: Change "Eastern part of the Andes" to "eastern Andes"

---

## Referee Comment (RC2) · Anonymous Referee #2 · 28 Dec 2018

The manuscript 'A Holocene black carbon ice-core record of biomass burning in the Amazon Basin from Illimani, Bolivia' provides a palaeo fire record from a region in desperate need of long-term records. The work offers an important contribution to our understanding of the relationship between palaeoclimate and the response of fire as important ecosystem driver. This manuscript is within the scope of 'Climate of the Past' and is well written and structured through the most part. With some minor changes to the writing style this manuscript can provide an important reference for palaeoclimatologists and palaeoecologists reconstructing the past of tropical South America.

The manuscript is mostly well written and structured, however, some sentences par-

ticularly in the Introduction and section 3.4 'Evidence of a Holocene Climatic Optimum dry period' could do with a closer examination. In the introduction the information is all there to set the scene, pose the questions and state the objectives but the interesting scope of this work is occasionally bogged down in redundant statements or overly long sentences. I would recommend a closer examination and edit of the introduction and section 3.4. by the authors.

Throughout the manuscript there is limited mention of non-biomass BC sources from major population centres within the Andes when covering the broken hockey-stick period, this needs to be expanded upon in the discussion if the concluding remark (P9/L32-33) is to stand.

Throughout the text vegetation burning in the Amazon Basin is identified as the source of rBC but little to no mention of more local burning of vegetation across the puna of the Altiplano or even the montane forests of the eastern Andean flank. Why is this not a reasonable source of at least part of the rBC signal?

The composite charcoal record for TSA (Fig 5e) records a noticeable drop around AD 1550-1600 corresponding to indigenous depopulation following European arrival. This decline is mirrored in the IL-99 rBC record (Fig 5a), is there a link? Also what is driving the increasing nitrate levels during this period as unlike the Industrial era increase its not linked to (NOx) traffic pollution and how is this linked to the decline in rBC? Also linked to this point, does the spike in the dust proxy (Ce) shortly following this (∼AD 1640) relate to historical changes in human population and land-use? This would seem to be an interesting point of discussion but is only briefly mentioned on P6/L32.

P2/L13 – Sentence reads '... burning was almost absent from the Amazon Basin before the 1960's'. It's a very big claim to suggest that fire was almost absent in Amazonia prior to the 60's. This point needs clarifying. Do you mean natural fires? Or are you suggesting that from pre-European arrival all the way through to the rubber boom people didn't clear and burn forest within the Amazon Basin?

[Figure]

The last paragraph of the discussion (P9/L9-15) brings up a fascinating change in the IL-99 record around 6000 BCE, which is speculated to correspond to the 8.2 k event in Greenland. Perhaps this is something that will be focused on in future work, however, expanding on this potentially controversial point and mentioning its signal or lack of in corresponding South American archives would be useful.

P1/L9 – suggest changing 'partially' to 'particularly'

P1/L23 – Remove 'an' from sentence '...dry period caused an exceptional biomass burning...'

P2/L19 – consider adding sensu (in terms of) to the reference (Marlon et al. 2008) if this is the first publication to pose the 'broken fire hockey stick hypothesis' and you are discussing your work in terms of this initial hypothesis.

P4/L3-8 – This is a 6 line sentence. Consider splitting in two or numbering the previous types of studies.

P8/L5-9 – Clarify this sentence so that wet conditions specifically are related to Bakker et al (2001) use of benthic/planktic diatoms to infer changes in water level and Lake Titicaca's overflowing and that colder conditions are linked to glacier advance (Zech et al. 2007).

P8/L6 – remove 'Late Glacial'. Dates provided and mention of Coipasa humid phase are sufficient.

P8/L9 – Remove 'then'

P8/L13 – Remove paragraph break from here.

P8/L12 – Comma after however

P8/L19 – If several studies show this reference the publications.

P8/L21 –This is the first time pollen is mentioned as a proxy. Is this related to the

reference as last sentence?

P8/L29 – Remove WD abbreviation if it's the only time used in the text.

P8/L32 – Here, consider changing 'opposite' to 'antiphase'.

P8/L39 – change to '. . .responsible for drier conditions. . .'

P9/L6 – Change 'forest extension' to 'forest expansion'

P9/L7-8 – Why is this occurring? Can you offer a suggestion or link to another record?

P9/L12 – Add in 'the' and remove end of sentence to read '. . .revealing that its impacts were also apparent in the southern South American tropics.'

Misc:

AD should go before date and BC after date e.g. AD 1730 / 1000 BC. Regardless change AD/BC to CE/BCE as suggested in COP house standards.

Add a comma to all numbers 10,000 and above.

Check throughout the manuscript for correct capitalization of geographical locations e.g. should be western/west not Western/West when not referring to specific place names.

NH is only used twice as an abbreviation for northern hemisphere, while southern hemisphere is written fully throughout. Suggest just abandoning the abbreviation.

Figure 2 – Perhaps due to conversion to PDF the lines on Fig 2a and c maybe denoting the change to the y-axis appear to have shrunk. This change should be clarified.

[Figure]

---

## Author Comment (AC1) · 8 Feb 2019

**Author's reply to peer-review comments on**

"A Holocene black carbon ice-core record of biomass burning in the Amazon Basin from Illimani, Bolivia", by Dimitri Osmont et al., submitted to CP.

We would like to thank the referees for the time they spent on our manuscript and for their constructive comments which helped us to improve the quality of this paper. Please find below our responses to your comments (in blue) and our changes to the manuscript (in red).
* * *
**Reviewer 1 (RC1, Anonymous Referee #1):**

"A Holocene black carbon ice-core record of biomass burning in the Amazon Basin from Illimani, Bolivia" presents a fire record from location and across a time frame where this information is sorely needed. This biomass burning record is an important contribution to fire science and to paleoclimatology. The well-written paper is within the scope of Climate of the Past, clearly presents the methodology and assumptions, and contains high-quality figures. It is evident that the authors carefully examined all aspects of the paper before submission, rather than trying to rush a draft through to submission, where the current presentation is well structured and clear. However, the authors should address the following concerns before publication:

We thank you for the overall positive evaluation and for the constructive comments and references provided. We considered all your comments and corrected the manuscript accordingly.

The authors assume (Page 3, Lines 8-9) that "Aerosol source apportionment studies in Amazonia have shown that recent BC emissions in this region originate only from biomass burning (Artaxo et al., 1998)". This previous statement pertains to Amazonia, and does not include the major urban areas of La Paz and El Alto which are located only 10s of kilometers from Illimani. The recent article "Black carbon emission and transport mechanisms to the free troposphere at the La Paz/El Alto (Bolivia) metropolitan area based on the Day of Census (2012)" clearly demonstrates the production of traffic-related BC and deposition on regional mountains (Wiedensohler et al., 2018). I realize that this article was published recently, and that the authors may not have known of its existence when they were writing their manuscript. However, as some of the authors of the Wiedensohler et al., 2018 paper collaborate with the authors of this Climate of the Past submission, it seems likely that the authors of the submitted paper may have known about this major source of BC. Previously-published literature also clearly demonstrates the effects of BC from urban centers on Andean glaciers (Schmitt et al., 2015). Although Schmitt et al. (2015) investigate glaciers in the Cordillera Blanca, these Peruvian glaciers still have the same main moisture source of the Amazon (and by extension, of the Atlantic) as Illimani. Molina et al. (2015 and references therein) mention that 50% of the BC in the Andean region is from biomass burning, which leaves the other 50% to be produced by other sources, of which fossil fuel burning (e.g. diesel-powered vehicles) is a major component. Therefore, the assumption that BC in Illimani after industrialization is only from biomass burning is does not accurately portray regional emissions.

We would like to thank the reviewer for providing interesting references which we were not aware of. The concern is justified, but we think that the major contribution to rBC in the IL-99 ice core is from biomass burning in the Amazon basin and not from fossil sources in La Paz. It does not mean that anthropogenic emissions did not contribute at all during the last decades, but that biomass burning was the predominant source. This is already mentioned in section 3.3: "we assume that anthropogenic BC emissions from fossil fuel combustion remain minor compared to biomass-burning-related BC emissions". Maybe our message was sometimes too firm and therefore we replaced expressions such as "only from biomass burning" by "predominantly from biomass burning" (in our conclusion). We also considered your references in our introduction and added: "However, in the Andean region, it has been shown that recent BC anthropogenic emissions from urban areas, particularly from traffic, could reach high elevations sites (Wiedensholer et al., 2018) and potentially affect local glacier melting (Molina et al., 2015; Schmitt et al., 2015).".
Our main argument is that rBC does not show the same increase as nitrate and Pb after 1960 (Fig. 3), which are good tracers for NOx from traffic emissions and leaded gasoline emissions, respectively (Eichler et al., 2015). Nitrate and Pb levels in the IL-99 ice core exponentially increase in the $2^{nd}$ half of the $20^{th}$ century due to traffic emissions while this is not the case for rBC (see Fig. 3 and also Fig. 2 for greater detail in the $20^{th}$ century). On the contrary, rBC and temperature present a very similar trend since 1730 CE (see discussion about correlation below). Maybe this is not obvious with 10-year averages in Fig. 3 so we attached below a figure comparing the IL-99 rBC, nitrate, Pb and temperature records at higher resolution. In Fig. 2, it is obvious that rBC variations

did not increase dramatically in the 2[nd] half of the 20[th] century as Pb and nitrate did. On the contrary, periods of higher concentrations (1900s, 1940s, 1960s) coincide with higher temperatures and lower precipitation in the south-western Amazon Basin, where industrial/traffic emissions are very limited. Of course we cannot disentangle between natural biomass burning and anthropogenic biomass burning (deforestation or agriculture). Wiedensholer et al. (2018) showed that the major contribution to rBC emissions around La Paz is due to traffic. If this were true for Illimani, our rBC record should show a trend similar to Pb or nitrate, as traffic is the dominant source for these two compounds in the last decades.

The study from Schmitt et al. (2015) presents some weaknesses which prevent a direct comparison with our IL-99 rBC record. First, they performed very few measurements above 6000 m except in region 2 (section 4.2). Second, they used a technique (LAHM) which does not discriminate between dust and BC (section 3). They admit that, in the case of region 2, most of the signal is due to dust and not BC (section 4.2). When using a SP2, they obtain very low rBC concentrations (0.65 ng g$^{-1}$) for region 2 (with "lower nearby population densities"), comparable to our values al Illimani, while for region 4, close to Huaraz, they obtain 50 times higher averages (31.0 ng g$^{-1}$), which illustrates the extent of the anthropogenic contamination.

It is to note that all the references provided by the reviewer discuss an anthropogenic impact only in the last years or decades, while our rBC concentrations steadily increase since 1730 CE, when no emissions could be attributed to traffic. Mining activities could also have influenced our rBC record, and this is widely discussed in section 3.3 and in our response to Reviewer 2.

The authors carry the above assumption through all of their work in section 3.2, and in section 3.3, page 7, lines 22-23. The authors do account for the possibility of a fossil fuel source for the BC from _1730 AD onwards (Page 7, Lines 24-37) including comparing their results to lead and nitrate concentrations in the IL-99 core from Eichler et al., 2015 where these concentrations reflect motor vehicle emissions. Comparing the rBC concentrations with the lead and nitrate records is a good idea, however, such a direct comparison may not be applicable. The authors only visually compare these records (Figure 5; second paragraph on page 7) and do not numerically investigate any correlations. The 10-year averages of lead drop after the switch to unleaded gasoline, while the 10-year nitrate averages continue to climb, which is similar to the increase in rBC (Figure 5). The sampling resolution is 10-cm samples, so in this uppermost section of the core, it should be possible to examine the data at a higher resolution than 10-year averages (as nicely demonstrated in Figure 2). What is the relationship between lead, nitrate and rBC over the past century when examined at a higher resolution? (As 10-year fixed averages are a rather arbitrary number, the decadal cut-off points may introduce errors when comparing the three records. E.g. one high value can inordinately influence an entire decadal mean). What happens if you compare the records with higher resolution moving averages from 1750 AD onward? If 50% of the BC is from biomass burning, then this source would moderate the rBC concentrations where you would not expect to see an equal rise in rBC as in lead and nitrate. Therefore, the conclusion (Page 9, lines 34 and 35) "Lastly, the rise in rBC concentrations since 1730 AM seems only driven by increased biomass burning levels due to higher temperatures and more intensive deforestation in the last decades but does not relate to fossil fuel rBC emissions" is misleading.

Please find below a figure showing the trend since 1730 CE for rBC, nitrate, Pb and ammonium, at higher resolution (annual) and using 11-year moving averages. The temperature reconstruction is available only a 10-year resolution but is based on the ammonium record (Kellerhals et al., 2010). We therefore used the ammonium record at annual resolution. Correlation coefficients are indicated. We also added in section 3.3 the correlation coefficient between the rBC and temperature 11-year moving averages ($r = 0.53$, $p < 0.05$, $n = 96$). We also compared the slope of the increase for rBC and temperature Z-scores, which is similar (see figure 2 below), and added this information in section 3.3 of the manuscript: "For the time period 1730–1999 CE, the rate of increase in concentration for both the rBC and temperature anomaly, obtained by linear regression of Z-scores calculated from 10-year means, is similar, with a slope of 0.011 yr$^{-1}$".

We therefore consider that our main conclusion remains unchanged but we agree that our message in the conclusion was too binary. We modified the sentence to say that rBC emissions from fossil fuels can represent a minor contribution and that biomass burning is not the only source.

Molina, L.T., Gallardo, L., Andrade, M., Baumgardner, D., Borbor-Córdova, M., Bórquez, R., Casassa, G., Cereceda-Balic, F., Dawidowski, L., Garreaud, R., Huneeus, N., Lambert, F., McCarty, J.L., Mc Phee, J., Mena-Carrasco, M., Raga, G.B., Schmitt, C., Schwarz, J.P., 2015. Pollution and its impacts on the South American cryosphere. Earth's Future, 3, 345-369, http://dx.doi.org/10.1002/2015EF000311.

Schmitt, C.G., All, J.D., Schwarz, J.P., Arnott, W.P., Cole, R.J., Lapham, E., Celestian, A., 2015. Measurements of light-absorbing particles on the glaciers in the Cordillera Blanca, Peru. Cryosphere 9, 331–340

Wiedensholer, A., Andrade, M., Weinhold, K., Mueller, T., Birmili, W., Velarde, F., Moreno, I., Forno, R., Sanchez, MF., Laj, P., Whiteman, D.N., Krejci, R., Sellegri, K., Reichler, T. (2018) Black carbon emission and transport mechanisms to the free troposphere at the La Paz/El Alto (Bolivia) metropolitan area based on the Day of Census (2012). Atmospheric Environment, 194, 158-169, DOI:https://doi.org/10.1016/j.atmosenv.2018.09.032

[Figure]

Figure 1: Comparison for the time period 1730-1999 AD for rBC, NH4, NO3 and EF Pb. Thin lines are annual averages, thick lines are 11-year moving averages. Correlation coefficients are between the respective species and rBC (in italic: based on annual averages; in bold: based on 11-year moving averages).

[Figure]

Figure 2: Comparison of slope (linear regression in red) of the increase in rBC and T anomaly since 1730 AD. Data are Z-scores of the 10 year averages.

Page 8, Lines 5 and 6: Higher lake levels in Lake Titicaca would certainly indicate a wetter climate, but why do you infer that these higher lake levels also depict a cooler climate? Due to less evaporation? The lake is also not "overflowing" when it has higher lake levels than present. Yes, the shorelines are higher than present, but the lake did not create continuous catastrophic floods.

Reviewer 2 also mentioned this point (please refer to the respective comment). The climate was wetter, as indicated by higher shorelines for Lake Titicaca (Baker et al., 2001), and colder at the same time, as indicated by Coipasa glacier advances (Zech et al., 2007) or the $\delta^{18}O$ ice-core records from Sajama (Thompson et al., 1998) and Illimani (Sigl et al., 2009). The main driver of Lake Titicaca level, as explained in Baker et al. (2001), is precipitation, inducing flow variations in the rivers feeding the lake. Evaporation probably plays a very minor role.
We always observe, throughout the Illimani record, that cold (warm) periods in tropical South America are characterized by wetter (drier) conditions, respectively. This is primarily due to ocean-atmosphere interactions and dynamics in this part of the globe.
We agree that the word "overflowing" sounds a bit tragic but it is used in the Baker paper. Here, we replaced it by "higher shorelines".

Figures 1 and 6: I realize that there are few paleofire records for this region and so it is difficult to find records with which to compare your rBC results. (This lack of records does nicely increase the value of your work). However, as charcoal only records local to semi-regional fires, Laguna La Gaiba is too far away to provide a useful comparison with the Illimani ice core record. Illimani is more of a regional record than any lake, but comparisons between individual records are limited by the spatial record of any individual record. The four lakes contain three different ecosystems (llanos, Amazon rain forest, and seasonally dry tropical forest) and so your decision to keep these records separate rather than making a composite record makes sense. However, the current comparison with Laguna La Gaiba is beyond the geographic limits of charcoal.

We agree with the statement that lake sediment charcoal records contain a local to semi-regional information and that Laguna La Gaiba is a bit far away from the Illimani site. However, seeing similarities between local signals suggests a larger scale signal, which is also present in the Illimani rBC record. We decided to keep the Laguna La Gaiba record for the following reasons:
-    Among the close records, it is the only one located in the seasonally dry tropical forest ecosystem.
-    Regarding the HCO, it shows a very similar trend to the Illimani rBC record, suggesting a consistent regional trend in the western Amazon Basin that is well reflected in the Illimani ice core.
-    The Illimani charcoal record (Brugger et al., in preparation) shows this similarity of amplified fire activity in the HCO that declined towards the late Holocene with several sites across the Bolivian lowland mentioned here, but not with sedimentary sites on the Altiplano nor with the Sajama charcoal record (Reese et al., 2013).
In order to warn the reader of the quite large distance between Illimani and Laguna La Gaiba, we added the following sentence: "although this site is located quite far from Illimani".

**Miscellaneous:**
Please use a comma in numbers with five places or more.

Done.

Page 1, Line 12: Place "the" before "Illimani".

Done.

Page 1, Line 22: Omit "an" before "exceptional biomass burning activity".

Done.

Page 1, Line 24: Place "in fire activity" or "in biomass burning" after "decrease". In the previous sentence you mention both increasing temperatures and deforestation. Therefore, it is necessary to re-define what is decreasing in the following sentence.

Done.

Page 3, Line 27: Change "boreholes" to "borehole". Even if you have multiple boreholes, the plural is already included in the word "temperatures".

Done.

Page 3, Line 27: Remove "the" before "Illimani".

You probably meant P.3 L.37. We instead replaced it by "The Illimani site".

Page 3, Line 27: Change "can receive moisture influx also during the dry season" to "can also receive moisture influx during the dry season".

You probably meant P.3 L.37. Done.

Page 5, Line 11: Change "peaking" to "peak".

Done. We also removed the word "changes".

Page 5, Line 12: Change "are not in agreement with" to "do not agree with".

Done.

Page 5, Line 21: Change "starting only" to "only starting".

Done.

Page 5, Line 27: Change "East" to "east".

Done.

Page 6, Line 13: Omit the word "explain" as the explanation is implicit in the word "contribute".

Done.

Page 6, line 16: Change "western part of the Andes" to "western Andes".

Done.

Page 6, Line 26: Replace "average" with "mean".

Done. We also replaced it throughout the paper when it was needed.

Page 7, Line 28: Replace "of" with "for".

Done.

Page 7, Line 30: Place "the" before "second half".

Done.

Page 7, Line 32: Change "charcoal composite records" to "composite charcoal records".

Done.

Page 7, Line 35: Change "could be" to "could also be".

Done.

Page 8, Line 39: Change "responsible of drier conditions" to "responsible for drier conditions".

Done.

Page 9, Line 23: Change "Eastern part of the Andes" to "eastern Andes".

Done.

**Reviewer 2 (RC2, Anonymous Referee #2):**

The manuscript 'A Holocene black carbon ice-core record of biomass burning in the Amazon Basin from Illimani, Bolivia' provides a palaeo fire record from a region in desperate need of long-term records. The work offers an important contribution to our understanding of the relationship between palaeoclimate and the response of fire as important ecosystem driver. This manuscript is within the scope of 'Climate of the Past' and is well written and structured through the most part. With some minor changes to the writing style this manuscript can provide an important reference for palaeoclimatologists and palaeoecologists reconstructing the past of tropical South America.

We acknowledge your positive evaluation and carefully took into account your suggestions. Please find our response below.

The manuscript is mostly well written and structured, however, some sentences particularly in the Introduction and section 3.4 'Evidence of a Holocene Climatic Optimum dry period' could do with a closer examination. In the introduction the information is all there to set the scene, pose the questions and state the objectives but the interesting scope of this work is occasionally bogged down in redundant statements or overly long sentences. I would recommend a closer examination and edit of the introduction and section 3.4. by the authors.

We had a detailed look at the manuscript and tried to simplify some sentences and to better organize the argumentation (e.g. in section 3.3), with a particular focus on sections 1 and 3.4. Please refer to the "track changes" version of the manuscript.

Throughout the manuscript there is limited mention of non-biomass BC sources from major population centres within the Andes when covering the broken hockey-stick period, this needs to be expanded upon in the discussion if the concluding remark (P9/L32-33) is to stand.

Reviewer 1 also mentioned this point in detail. Please refer to the corresponding response to Reviewer 1.

Throughout the text vegetation burning in the Amazon Basin is identified as the source of rBC but little to no mention of more local burning of vegetation across the puna of the Altiplano or even the montane forests of the eastern Andean flank. Why is this not a reasonable source of at least part of the rBC signal?

We cannot fully exclude some influence. However, it is impossible to disentangle the origin of rBC without additional analyses of the ice core (e.g. pollen, which enables to indicate the predominant source region based on vegetation types). Since the Amazon Basin is a much larger source, we assume it dominates the signal. Compared to the Bolivian lowlands, burning across the Altiplano is very limited due to the scarcity of vegetation cover. Moreover, our correlation analyses with temperature and precipitation (fig. 4) clearly show that the rBC record only correlates with regions located east, in the Amazon Basin, and not with the Altiplano. Regarding the montane forests of the eastern Andean flank, we implicitly included them in the Amazon Basin as both are located east of Illimani and we cannot disentangle them with our rBC analyses.

The composite charcoal record for TSA (Fig 5e) records a noticeable drop around AD 1550-1600 corresponding to indigenous depopulation following European arrival. This decline is mirrored in the IL-99 rBC record (Fig 5a), is there a link? Also what is driving the increasing nitrate levels during this period as unlike the Industrial era increase its not linked to (NOx) traffic pollution and how is this linked to the decline in rBC? Also linked to this point, does the spike in the dust proxy (Ce) shortly following this (~AD 1640) relate to historical changes in human population and land-use? This would seem to be an interesting point of discussion but is only briefly mentioned on P6/L32.

The composite charcoal record for TSA and the rBC record agree well at this time period, with two local maxima around AD 1500 and 1650, and a drop around AD 1550–1600, superimposed on an overall declining trend. It thus seems reasonable to say that there is a link between the 2 drops. Our assumption is that the overall declining trend could refer to the climatic signal (transition from a dry/warm MWP to a cold/wet LIA, leading to a decline in fire activity), while small superimposed variations could potentially reflect an anthropogenic origin, due to human-induced fires and mining activities implying wood burning for ore smelting in furnaces.
Nitrate is produced by high-temperature combustion processes, reflecting either mining activities or biomass burning as suggested by Kellerhals et al. (2010).
We did not discuss the AD 1550–1600 drop but we thank the reviewer for bringing an interesting hypothesis. This decline could be partially explained by the depopulation following European arrival, inducing a decline in

human-induced fires and mining activities. However, neither the nitrate record nor the Pb record (Eichler et al., 2015) show a clear decline at that time. Mining activities were not stopped after the beginning of the Spanish colonization in AD 1532 but, on the contrary, they rapidly increased, especially after the discovery of the huge silver deposit in Potosi in AD 1545 (Eichler et al., 2015). The decline in the anthropogenic Pb record of Illimani after AD 1570 (and until the mid-17[th] century) is best explained by a technical evolution in smelting processes due to the introduction of the amalgamation process requiring less fuel (Eichler et al., 2015). It is therefore possible that the decline in rBC also reflects this technological evolution as less wood was burned.

According to Power et al. (2012), the post-AD 1500 composite charcoal record for TSA does not clearly reflect climate change or demographic collapse due to the large climatic, topographic and vegetation differences encompassed in this region. At the scale of the Americas, the same study concluded that the LIA climatic change was more important than the demographic collapse to explain the post-AD 1500 biomass burning decline in Americas.

To our knowledge, we are not able to convincingly explain the Ce peak around AD 1600–1650 regarding climatic implications. In the Illimani ice core, some sections with poor ice quality (brittle ice) are found around AD 1595–1620 and 1625–1640, which can be critical for trace element analyses (such as Ce) as contamination is a risk. We removed the mention of this peak as a possible dry period and added the following sentence in the caption of fig. 5: "peak values from 1610 to 1630 CE might be due to poor ice quality prone to contamination".

We reformulated our argumentation to take into account the aforementioned discussion. Some sentences were moved. We kindly ask the reviewer to refer to the first two paragraphs of section 3.3.

P2/L13 – Sentence reads '…burning was almost absent from the Amazon Basin before the 1960's'. It's a very big claim to suggest that fire was almost absent in Amazonia prior to the 60's. This point needs clarifying. Do you mean natural fires? Or are you suggesting that from pre-European arrival all the way through to the rubber boom people didn't clear and burn forest within the Amazon Basin?

This is what is mentioned in the fire inventory from Mouillot and Field (2005) (P.407). For clarity, we moved the reference to the end of the sentence.

In tropical rain forests at the center of the Amazon Basin, natural fires are very rare due to the very humid climate (Bowman et al., 2011; Cochrane, 2003). In this remote and inaccessible region, deforestation (mainly by means of fires) started in the 1960s and became significant only between 1975 and 1980 (Mouillot and Field, 2005). It started from the southern and eastern edges of the Amazon Basin, known as the "arc of deforestation" and followed the creation of roads.

The last paragraph of the discussion (P9/L9-15) brings up a fascinating change in the IL-99 record around 6000 BCE, which is speculated to correspond to the 8.2 k event in Greenland. Perhaps this is something that will be focused on in future work, however, expanding on this potentially controversial point and mentioning its signal or lack of in corresponding South American archives would be useful.

We think that expanding on this topic is outside the scope of this paper. This is a different and single topic by itself. Here, we can just speculate about this feature but we do not have enough evidence in our record to make a clear statement.

P1/L9 – suggest changing 'partially' to 'particularly'.

Done.

P1/L23 – Remove 'an' from sentence '…dry period caused an exceptional biomass burning…'.

Done.

P2/L19 – consider adding sensu (in terms of) to the reference (Marlon et al. 2008) if this is the first publication to pose the 'broken fire hockey stick hypothesis' and you are discussing your work in terms of this initial hypothesis.

After careful search, we realized that the expression "broken fire hockey stick" had never been used in a scientific publication before but was only informally mentioned by the community. Therefore, we decided to remove this expression from our paper.

P4/L3-8 – This is a 6 line sentence. Consider splitting in two or numbering the previous types of studies.

We split this sentence. It now reads: "The IL-99 ice core has already been widely studied. Knüsel et al. (2005) investigated the potential impact of ENSO on the ionic records, revealing elevated dust values during warm phases of ENSO. Kellerhals et al. (2010b) used thallium as a possible volcanic eruption tracer. Kellerhals et al. (2010a) produced a regional temperature reconstruction for the last 1600 years based on the $NH_4^+$ record. Eichler et al. (2015) focused on the historical reconstruction of regional silver production and recent leaded gasoline pollution based on the lead record. Finally, Eichler et al. (2017) made use of the copper record to reconstruct copper metallurgy, showing that earliest extensive copper metallurgy started in the Andes 2700 years ago."

P8/L5-9 – Clarify this sentence so that wet conditions specifically are related to Bakker et al (2001) use of benthic/planktic diatoms to infer changes in water level and Lake Titicaca's overflowing and that colder conditions are linked to glacier advance (Zech et al. 2007).

According to Zech et al. (2007), Coipasa glacier advances are the result of both wetter and colder conditions in this region.
We clarified this sentence. This point was also raised by reviewer 1. It now reads: "Over the Bolivian Altiplano, wet conditions were evidenced by higher shorelines of Lake Titicaca between 11,000 and 9500 BCE (Baker et al., 2001), inferred from benthic/planktonic diatom fractions, while cold conditions were suggested by glacier advances in the Cordillera Real between 11,000 and 9000 BCE during the "Coipasa" humid phase (Zech et al., 2007). This illustrates that the Younger Dryas was not dry on a global scale despite increasing dustiness in Greenland ice-core records (Mayewski et al., 1993)."
We also provided additional details about this topic in the caption of fig. 6: "A higher percentage indicates a lower lake level (drier conditions) as salinity increases in the lake, leading to the precipitation of $CaCO_3$ and its deposition in the sediments".

P8/L6 – remove 'Late Glacial'. Dates provided and mention of Coipasa humid phase are sufficient.

Removed.

P8/L9 – Remove 'then'.

Removed.

P8/L13 – Remove paragraph break from here.

Removed.

P8/L12 – Comma after however.

Added.

P8/L19 – If several studies show this reference the publications.

These are the references listed below in the same paragraph. We do not think that it is necessary to repeat here the 8 references already mentioned in this paragraph. We also replaced "several other" by "numerous".

P8/L21 –This is the first time pollen is mentioned as a proxy. Is this related to the reference as last sentence?

Yes it is. We added "from the same record".

P8/L29 – Remove WD abbreviation if it's the only time used in the text.

We removed it.

P8/L32 – Here, consider changing 'opposite' to 'antiphase'.

Done.

P8/L39 – change to '…responsible for drier conditions…'.

Done.

P9/L6 – Change 'forest extension' to 'forest expansion'.

Done.

P9/L7-8 – Why is this occurring? Can you offer a suggestion or link to another record?

The rBC variations are in line with the temperature reconstruction from the Illimani ice core from Kellerhals et al. (2010). Slightly lower temperatures (fig. 7 in the aforementioned paper) were observed roughly between 700 and 1000 CE. On the contrary, temperatures were slightly higher before 700 CE. Unfortunately, this reconstruction does not go beyond 350 CE. We therefore added in the manuscript the following sentence: "This in line with the temperature reconstruction from Illimani showing slightly lower temperatures between approximately 700 and 1000 CE compared to the time period before (Kellerhals et al., 2010a)".

P9/L12 – Add in 'the' and remove end of sentence to read '…revealing that its impacts were also apparent in the southern South American tropics.'

Done.

**Misc:**
AD should go before date and BC after date e.g. AD 1730 / 1000 BC. Regardless change AD/BC to CE/BCE as suggested in COP house standards.

We replaced AD/BC by CE/BCE. We also made the changes in the figures.

Add a comma to all numbers 10,000 and above.

Done.

Check throughout the manuscript for correct capitalization of geographical locations e.g. should be western/west not Western/West when not referring to specific place names.

Done.

NH is only used twice as an abbreviation for northern hemisphere, while southern hemisphere is written fully throughout. Suggest just abandoning the abbreviation.

We abandoned the abbreviation.

Figure 2 – Perhaps due to conversion to PDF the lines on Fig 2a and c maybe denoting the change to the y-axis appear to have shrunk. This change should be clarified.

Maybe this results from the final conversion of the manuscript to PDF format. However, I do not see this feature on the original PDF figure generated from the plot. As this is this figure that will be finally submitted separately from the text at the end of the submission process, this does not pose a problem.

**References**

Baker, P. A., Seltzer, G. O., Fritz, S. C., Dunbar, R. B., Grove, M. J., Tapia, P. M., Cross, S. L., Rowe, H. D., and Broda, J.P.: The history of South American tropical precipitation for the past 25 000 years, Science, 291, 640–643, 2001.

Bowman, D. M. J. S., Balch, J., Artaxo, P., Bond, W. J., Cochrane, M. A., D'Antonio, C. M., DeFries, R., Johnston, F. H., Keeley, J. E., Krawchuk, M. A., Kull, C. A., Mack, M., Moritz, M. A., Pyne, S., Roos, C. I., Scott, A. C., Sodhi, N. S., and Swetnam, T. W.: The human dimension of fire regimes on Earth, J. Biogeogr., 38, 2223–2236, 2011.

Brugger, S. O., Gobet, E., Osmont, D., Behling, H., Fontana, S. L., Hooghiemstra, H., Morales-Molino, C., Sigl, M., Schwikowski, M., and Tinner, W.: Tropical Andean glacier reveals Colonial legacy in modern montane ecosystems, in preparation.

Cochrane, M. A.: Fire science for rainforests, Nature, 421, 913–919, 2003.

Eichler, A., Gramlich, G., Kellerhals, T., Tobler, L., and Schwikowski, M.: Pb pollution from leaded gasoline in South America in the context of a 2000-year metallurgic history, Sci. Adv., 1, e1400196, 2015.

Kellerhals, T., Brütsch, S., Sigl, M., Knüsel, S., Gäggeler, H. W., and Schwikowski, M.: Ammonium concentration in ice cores: A new proxy for regional temperature reconstruction?, J. Geophys. Res., 115, D16123, 2010.

Mouillot, F. and Field, C.: Fire history and the global carbon budget: a $1° × 1°$ fire history reconstruction for the 20th century, Global Change Biol., 11, 398–420, 2005.

Power, M. J., Mayle, F. E., Bartlein, P. J., Marlon, J. R., Anderson, R. S., Behling, H., Brown, K. J., Carcaillet, C., Colombaroli, D., Gavin, D. G., Hallett, D. J., Horn, S. P., Kennedy, L. M., Lane, C. S., Long, C. J., Moreno, P. I., Paitre, C., Robinson, G., Taylor, Z., and Walsh, M. K.: Climatic control of the biomass-burning decline in the Americas after AD 1500, Holocene, 23, 3–13, 2012.

Reese, C. A., Liu, K. B., and Thompson, L. G.: An ice-core pollen record showing vegetation response to Late-glacial and Holocene climate changes at Nevado Sajama, Bolivia, Ann. Glaciol., 54(63), 183–190, 2013.

Schmitt, C. G., All, J. D., Schwarz, J. P., Arnott, W. P., Cole, R. J., Lapham, E., and Celestian, A.: Measurements of light-absorbing particles on the glaciers in the Cordillera Blanca, Peru, Cryosphere, 9, 331–340, 2015.

Sigl, M., Jenk, T. M., Kellerhals, T., Szidat, S., Gäggeler, H. W., Wacker, L., Synal, H.-A., Boutron, C., Barbante, C., Gabrieli, J., and Schwikowski, M.: Instruments and Methods Towards radiocarbon dating of ice cores, J. Glaciol., 55, 985–996, 2009.

Thompson, L. G., Davis, M. E., Mosley-Thompson, E., Sowers, T. A., Henderson, K. A., Zagorodnov, V. S., Lin, P.-N., Mikhalenko, V. N., Campen, R. K., Bolzan, J. F., Cole-Dai, J., and Francou, B.: A 25,000-year tropical climate history from Bolivian ice cores, Science, 282, 1858–1864, 1998.

Wiedensholer, A., Andrade, M., Weinhold, K., Mueller, T., Birmili, W., Velarde, F., Moreno, I., Forno, R., Sanchez, M. F., Laj, P., Whiteman, D. N., Krejci, R., Sellegri, K., and Reichler, T.: Black carbon emission and transport mechanisms to the free troposphere at the La Paz/El Alto (Bolivia) metropolitan area based on the Day of Census (2012), Atmos. Environ., 194, 158–169, 2018.

Zech, R., Kull, C., Kubik, P. W., and Veit, H.: LGM and Late Glacial glacier advances in the Cordillera Real and Cochabamba (Bolivia) deduced from [10]Be surface exposure dating, Clim. Past, 3(4), 623–635, 2007.